# Magnetic Stirring Assisted Demulsification Dispersive Liquid–Liquid Microextraction for Preconcentration of Polycyclic Aromatic Hydrocarbons in Grilled Pork Samples

**DOI:** 10.3390/toxics7010008

**Published:** 2019-02-13

**Authors:** Jitlada Vichapong, Yanawath Santaladchaiyakit, Rodjana Burakham, Supalax Srijaranai

**Affiliations:** 1Creative Chemistry and Innovation Research Unit, Department of Chemistry and Center of Excellence for Innovation in Chemistry, Faculty of Science, Mahasarakham University, Maha Sarakham 44150, Thailand; 2Department of Chemistry, Faculty of Engineering, Rajamangala University of Technology Isan, Khon Kaen Campus, Khon Kaen 40000, Thailand; sanyanawa@gmail.com; 3Materials Chemistry Research Center, Department of Chemistry and Center of Excellence for Innovation in Chemistry, Faculty of Science, Khon Kaen University, Khon Kaen 40002, Thailand; rodjbu@kku.ac.th (R.B.); supalax@kku.ac.th (S.S.)

**Keywords:** magnetic stirring assisted demulsification dispersive liquid–liquid microextraction, polycyclic aromatic hydrocarbons, grilled pork, high performance liquid chromatography

## Abstract

A simple microextraction method, magnetic stirring assisted demulsification dispersive liquid–liquid microextraction, for preconcentration of five polycyclic aromatic hydrocarbons (fluorene, phenanthrene, anthracene, fluoranthrene, and pyrene) was investigated prior to analysis by high performance liquid chromatography. In this method, a mixture of extraction solvent and disperser solvent was rapidly injected into sample solution. The magnetic stirrer agitator aided the dispersion of the extraction solvent into the sample solution. After the formation of an emulsion, the demulsifier was added, resulting in the rapid separation of the mixture into two phases. No centrifugation step was required. Several parameters affecting the extraction efficiency of the proposed method were studied, including addition of salt, kind and volume of extraction solvent, volume of demulsifier solvent, and extraction times. Under the optimum conditions, high enrichment factor, low limit of detections (LODs) and good precision were gained. The proposed method was successfully applied to analysis of polycyclic aromatic hydrocarbon residues in grilled pork samples.

## 1. Introduction

Polycyclic aromatic hydrocarbons (PAHs) constitute of a large class of organic materials which are formed of two or more fused aromatic rings [1,2,3]. These compounds are generally drawn from incomplete combustion or high-temperature pyrolysis of organic materials, such as coal, petrol, wood, garbage, tobacco, meats or other organic foods [4]. Both the European Union (EU) and the United States Environmental Protection Agency (USEPA) have their own list of 16 PAHs as “priority organic pollutants”, due to their high toxicity to the human health [5]. PAHs are non-polar, very hydrophobic compounds, with low water solubility. Therefore, in the aquatic environment they commonly exist in relatively low concentrations [6]. Consequently, powerful analytical methods are required to extract, separate, and identify these target analytes in the environment [7].

Several chromatographic methods including gas chromatography [8,9] and high-performance liquid chromatography [10,11] have been regularly used for separation and quantification of PAHs in various samples. Although these are sensitive and selective methods, PAHs usually occur in ppb level or lower in complex mediums containing various interfering compounds [12]. Therefore, sample preparation, matrix removal and preconcentration of the target analytes are needed before analysis in order to obtain sensitive and accurate results.

Conventional sample preparation methods such as solid-phase extraction (SPE) [13,14] and liquid–liquid extraction (LLE) [15,16] have been used for preconcentration and clean-up before analysis of PAHs. Unfortunately, these methods are tedious, time-consuming, and require large amounts of samples and toxic organic solvents [2]. Therefore, much effort has been made to develop a simple, sensitive and environmentally friendly sample preparation method termed dispersive liquid–liquid microextraction (DLLME) [17]. This method is based on the use of high density of extraction solvents such as chlorinated solvent. However, these are toxic and harmful to human health and environment. The other DLLME mode is based on the low density of extraction solvent. However, the main drawback of DLLME modes was the requirement of a centrifugation step. Recently, a new DLLME technique without a centrifugation step was introduced, the low-density based demulsification DLLME [18,19]. After the mixture of extraction solvent and disperser solvent was added into the aqueous solution, the cloudy solution was immediately separated by adding demulsifiers. No centrifugation step was required.

There are various agitators, such as vortex [20], ultrasound [21], and in-syringe [22], that have been used to enhance the dispersion and to accelerate the formation of fine droplets of extraction solvent. Zhang et al. [23] proposed a simple magnetic stirring assisted dispersive liquid–liquid microextraction (MSA-DLLME) method. After the injection of extraction solvent and disperser solvent into an aqueous solution, the sample was magnetically stirred. Consequently, the centrifugal step was not required. In this present study, the magnetic stirring assisted demulsified dispersive liquid–liquid microextraction combined with HPLC was investigated for extraction, preconcentration and simultaneous determination of PAHs. The magnetic stirrer was used to increase the dispersion and mass transfer between two phases. Special attention was paid to parameters providing the highest extraction efficacy of the extraction procedure. We carefully evaluated salt addition, kind and volume of extraction solvent, kind and volume of disperser solvent, volume of demulsifier solvent and extraction times. The proposed method was successfully applied to the determination of PAHs in grilled pork samples. To our knowledge, this was the first time that the simple magnetic stirring assisted demulsified dispersive liquid–liquid microextraction technique was used in the determination of trace levels of PAHs.

## 2. Materials and Methods

### 2.1. Chemicals and Reagents

All chemicals and reagents used were of analytical reagent grade or better. The analytical standards of polycyclic aromatic hydrocarbon including fluorene, phenanthrene, anthracene, fluoranthrene and pyrene were purchased from Sigma-Aldrich (Darmstadt, Germany). The stock standard solution (1000 mg·L^−1^) was prepared by dissolving in methanol (MeOH). The working solution of standard was prepared daily by dilution with deionized water. Deionized water was prepared using RiOs^TM^ Type I Simplicity 185 (Millipore Waters, Newford, MA, USA) with the resistivity of 18.2 MΩ.cm. Acetonitrile (ACN) and MeOH (HPLC grade) were obtained from Merck (Darmstadt, Germany). All solvents for HPLC were filtered through 0.45 μm filters (Millipore Corp., Newford, MA, USA) and degassed in an ultrasonic bath. NaCl, anhydrous Na_2_CO_3_ and anhydrous Na_2_SO_4_ were purchased from Ajax Finechem (Auckland, New Zealand), and CH_3_COONa (Carlo Erba, France) was used. 1-Dodecanol and 1-octanol were provided by Merck (Darmstadt, Germany).

### 2.2. Chromatographic Conditions

The HPLC system was comprised of a Waters 1525 binary HPLC pump (Newford, MA, USA), a Rheodyne injector and photodiode array detector (PDA). The Empower 3 software was chosen for data acquisition. A LiChrospher^®^ 100 RP-8 endcapped (4.6 × 150 mm, 5.0 µm) column (Merck, Darmstadt, Germany) was employed for the separation of all analytes, and was maintained at room temperature. The mobile phase was comprised of 67% acetonitrile in water with isocratic elution at a flow rate of 1 mL·min^−1^. The injection volume was 20 µL. The detection of all target analytes was set at 254 nm. Five polycyclic aromatic hydrocarbons were separated within 10 min with the elution order of fluorene (*t_R_* = 6.87 min), phenanthrene (*t_R_* = 7.03 min), anthracene (*t_R_* = 7.98 min), fluoranthrene (*t_R_* = 8.98 min) and pyrene (*t_R_* = 9.46 min).

### 2.3. Sample Preparation of Grilled Pork Samples

Grilled pork samples were collected from market in Maha Sarakham province, Northeastern Thailand. One gram of spiked and non-spiked were weighed and transferred to a centrifuge tube. Then, 10 mL of extraction solvent (2 mol·L^−1^ KOH in ethanol) was added to hydrolyze the sample. This sample was ultrasounded with fixed power for 5 min at 45 °C, followed by centrifugation (15 min, 4,000 rpm). The supernatant was collected into another centrifuge tube. The solid residue was extracted again with 3 mL of extraction solvent. All volumes of supernatant were gathered, the funnel was frozen for 1 h at −18 °C to precipitate the fat of the solution. Approximately 5 mL of solution were kept and followed by centrifugation. After that, an aliquot was applied for magnetic stirring assisted demulsified dispersive liquid–liquid microextraction under the selected conditions and analysis by a HPLC system.

### 2.4. Magnetic Stirring Assisted Demulsified Dispersive Liquid–Liquid Microextraction

A 10-mL aliquot of standard solution of each polycyclic aromatic hydrocarbon or grilled pork sample was mixed with 20% (*w*/*v*) of NaCl and then mixture solution containing 75 µL of extraction solvent (1-dodecanol) and 500 µL dispersive solvent (ACN) was quickly injected into the sample solution. The solution was then stirred at 1500 rpm to increase the mass transfer between two phases. An emulsion (water/extraction solvent/dispersive solvent) was formed. After that, 500 µL of ACN (as de-emulsifier solvent) was injected into the solution to break down the emulsion. The extraction was found to float to the top of the solution immediately. The extraction phase was kept and injected to HPLC for analysis.

### 2.5. Method Validation

The method validations such as linearity of calibration graph, limit of detection (LOD), limit of quantitation (LOQ), precision and enrichment factors (EFs) were studied. The linear range of standard calibration was conducted between 0.0005 and 1 mg·L^−1^ of PAH. LODs and LOQs were defined as the lowest detectable concentration with a signal to noise ratio of 3:1 and 10:1, respectively. EFs were defined as the concentration ratio of the analytes in the settled phase after performing microextraction methods and its initial concentration in the aqueous phase. The relative standard deviation (RSD, %) for intra-days and inter-day of extraction of PAHs were determined at three different concentration levels (0.10, 0.25 and 0.35 mg·L^−1^ of each PAHs) using 5 injections. The enrichment factor (EF) was defined as the concentration ratio of the analytes in the settled phase (Cset) and in the aqueous sample (Co).

### 2.6. Statistical Analysis

Data results are given as the mean ± standard deviation (SD) of three measurements (*n* = 3). In all graphs, a linear regression analysis was conducted using Microsoft Excel 2013 software (London, EC1V 2NX, United Kingdom).

## 3. Results and Discussion

### 3.1. Optimization of Magnetic Stirring Assisted Demulsified Dispersive Liquid–Liquid Microextraction Condition

In order to improve monitoring of PAH using magnetic stirring assisted demulsified dispersive liquid–liquid microextraction coupled with HPLC analysis, various experimental parameters including addition of salt, kind and volume of extraction solvent, volume of demulsifier solvent and extraction times were evaluated. The aqueous solution (10.00 mL) containing 100 ng·mL^−1^ of each PAH was used for optimization. All the experiments were performed in triplicate and the mean of the results were used for optimization.

Generally, the addition of salt decreases the solubility of analytes in aqueous solution [24,25] and therefore increases their partitioning into the organic phase for liquid phase microextraction. In this study the addition of various salts (1.0 g) such as sodium chloride (NaCl), sodium sulphate (Na_2_SO_4_), sodium carbonate (Na_2_CO_3_), sodium acetate (CH_3_COONa) were considered and compared to a control, without salt addition. A comparison of extraction efficiency in terms of peak area with various kinds of salt is shown in Figure 1. Without salt addition, phase separation did not complete. The results demonstrated that the addition of NaCl and Na_2_SO_4_ provided no significant difference in the peak area of all PAHs except fluorene and phenanthrene. Therefore, NaCl was added in varying concentrations for further study. The effects of the amount of NaCl range from 0.5 to 2.5 g on the extraction efficiency of target analytes were also studied. The result (shown in Figure 2), was that the extraction efficacy of the analytes slightly increased as NaCl increased from 0.5 to 2.0 g, and trended to be decrease when the amount of NaCl was greater than 2.0 g. Finally, 2.0 g was chosen as the amount of NaCl in following experiments.

Choosing a suitable extraction solvent is important for obtaining an efficient extraction procedure [26]. In this study, low density immiscible solvents (density = *d*) were investigated, 1-octanol (*d* = 0.8270 g·mL^−1^), 1-undecanol (*d* = 0.8298 g·mL^−1^), and 1-dodecanol (*d* = 0.8309 g·mL^−1^). The cloudy solution was observed after a quick injection of a mixture of extraction solvent and dispersive solvent into the sample solution, the solution was then mixed using the magnetic stirrer. While using 1-octanol as an extraction solvent, it was found that phase separation did not occur. Thus, it was not suitable to be used for extraction solvent. In comparison of extraction efficacy between 1-undecanol and 1-dodecanol, it was found that 1-dodecanol provided high extraction efficacy in terms of peak area (data not shown). Thus, 1-dodecanol was chosen as the extraction solvent for further study. The effect of the 1-dodecanol volume was investigated in the range of 50–150 µL. It was found that, 50 µL of 1-dodecanol cannot complete phase separation. The peak area of all the target compounds decreased as the extraction solvent increased beyond 75 µL. Consequently, 1-dodecanol 75 µL was used as an optimum extraction solvent.

The selection of disperser solvent is important parameter in DLLME technique. This solvent should be miscible in both phases (extraction solvent and aqueous phase), moreover it should disperse the extraction solvent into the aqueous solution to form a cloudy state. To simplify the choosing process, the dispersive solvent should also be used as the demulsifier to break up the oil-in-water emulsion [27]. Various kinds of solvent such as ethanol, acetone, acetonitrile and methanol, were studied. The results are shown in Figure 3. It was found that acetonitrile as the disperser and demulsified solvent provided highest peak area. Thus, acetonitrile was chosen as disperser and demulsified solvent.

The volume of disperser solvent was studied within the range of 250–1,000 µL. It was found that the extraction efficiency slightly increased with increasing the volume of disperser solvent up to 500 µL and then decreased (as shown in Figure 4). Therefore, 500 µL of acetonitrile was used as disperser solvent. Moreover, volume of demulsified solvent was varied in the range 250–1,250 µL. It can be observed that a volume of 500 µL of demulsified solvent provided highest extraction efficiency in terms of peak area (as shown in Figure 5). Thus, acetonitrile 500 µL was selected as demulsified solvent.

In the two immiscible phase system, the mass transfer between the aqueous solution and extraction solvent depended on the extraction time and agitation [28]. The extraction time was defined as time that the sample was magnetic stirring agitated [29]. Extraction times on the extraction efficiency were evaluated for 1–10 minutes (data not shown) at 1500 rpm. The maximum peak area with the least standard deviation was obtained at four minutes, therefore it was selected as the optimum extraction time.

In summary, the optimum extraction conditions were sample solution 10.00 mL, 2.0 g NaCl, 1-dodecanol 75 µL used as extraction solvent, acetonitrile used as disperser (500 µL) and de-emulsified solvent (500 µL), and an extraction time of four minutes.

### 3.2. Analytical Performance of the Proposed Method

The analytical performances of the proposed method for analysis of polycyclic aromatic hydrocarbons were validated by obtaining linear range, coefficient of determination (*R*^2^), Limit of detections (LODs), limit of quantitations (LOQs,) and enrichment factors (EFs). Table 1 summarizes the analytical performance of the proposed method. LODs and LOQs ranged from 0.0001 to 0.0003 mg·L^−1^ and 0.0003 to 0.0005 mg·L^−1^, respectively. The calibration curve exhibited linearity over the range of 0.0005–1 mg·L^−1^ with *R*^2^ greater than 0.99. RSDs were in the range of 0.22–1.48% and 2.73–7.22% for retention time and peak area, respectively. High EFs (60–67) were also gained. Figure 6 depicts typical chromatogram comparing analyses of (a) a mixture standard of polycyclic aromatic hydrocarbons with direct injection by HPLC and (b) a mixture of standard of polycyclic aromatic hydrocarbons after magnetic stirring assisted demulsified dispersive liquid–liquid microextraction technique.

### 3.3. Grilled Pork Analysis

To eliminate the matrix effect in real sample analysis, matrix-match calibration was used. A high degree of linearity was observed in the range of 0.01–0.5 mg·g^−1^ with *R*^2^ greater than 0.995. LODs of the target analytes in real samples were studied with 3.3 × (SD y-Intercept/Average Slope) and LOQs were investigated using 10 × (SD y-Intercept/Average Slope) [24]. The obtained LODs and LOQs were in the range 0.001–0.005 mg·kg^−1^ and 0.004–0.010 mg·kg^−1^, respectively. These are below the established EU maximum level (12 µg·kg^−1^). For this analysis, all studied polycyclic aromatic hydrocarbons were found in the range of 0.30–1.00 mg·kg^−1^. The results are summarized in Table 2. To test the accuracy of the proposed method, recovery was investigated by spiking standard solution at different levels (0.01, 0.05, 0.10 mg·kg^−1^) before magnetic stirring assisted demulsified dispersive liquid–liquid microextraction. Figure 7 shows the chromatograms of grilled pork sample (untreated) (Figure 7a) as well as grilled pork samples spiked at concentrations of 0.01 mg·kg^−1^ (Figure 7b), 0.05 mg·kg^−1^ (Figure 7c), and 0.10 mg·kg^−1^ (Figure 7d). The chromatograms show that the sensitivity increases with increasing PAH concentration. The results (Table 3) show good analytical performance of the proposed method with the average recoveries for all studied analytes ranging from 82% to 99%, and good precision with relative standard deviation (RSD) less than 9%.

## 4. Conclusions

A fast and simple combining apparatus magnetic stirring assisted demulsified dispersive liquid–liquid microextraction has been investigated for extraction and preconcentration of polycyclic aromatic hydrocarbon residues coupled to high-performance liquid chromatographic analysis. In this procedure, a magnetic stirring-assisted process was used to accelerate the formation of fine droplets, which increased the extraction efficacy and decreased the extraction time. No centrifugation step was required. High enrichment factors, low LODs and good repeatability were obtained. The proposed method is both easy and rapid. As it uses less toxic organic solvent, it is also an environmentally friendly technique for the determination of polycyclic aromatic hydrocarbons in grilled pork samples.

## Figures and Tables

**Figure 1 toxics-07-00008-f001:**
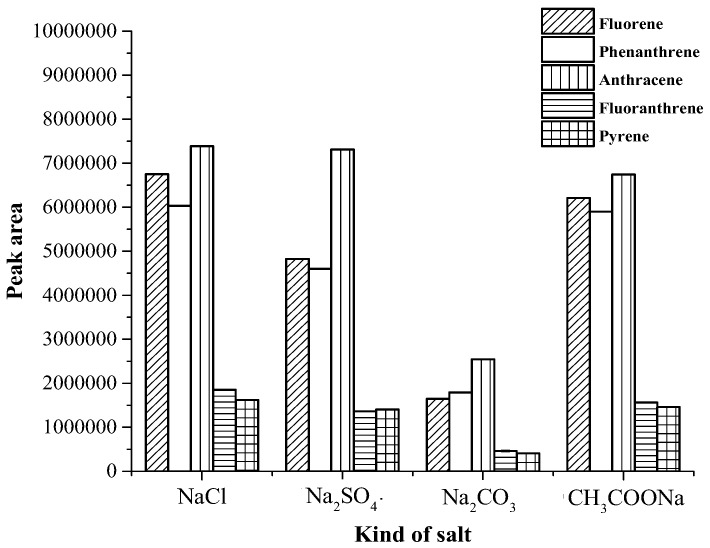
Effect of addition of salt on the extraction efficacy.

**Figure 2 toxics-07-00008-f002:**
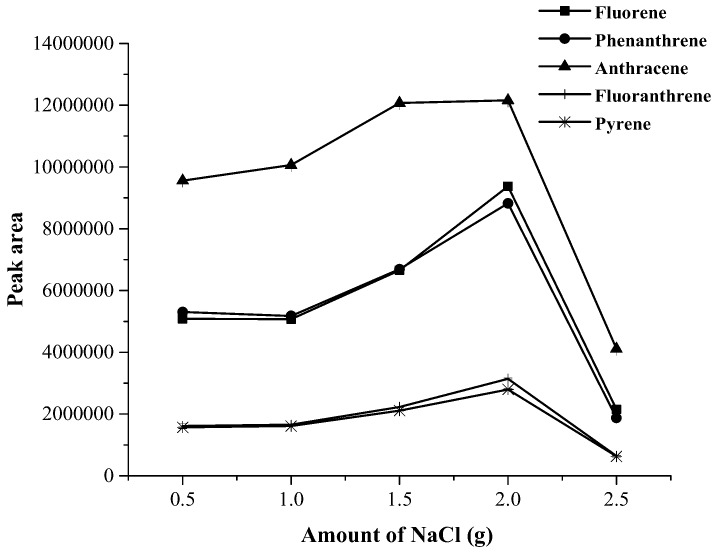
Effect of NaCl amount on the extraction efficacy.

**Figure 3 toxics-07-00008-f003:**
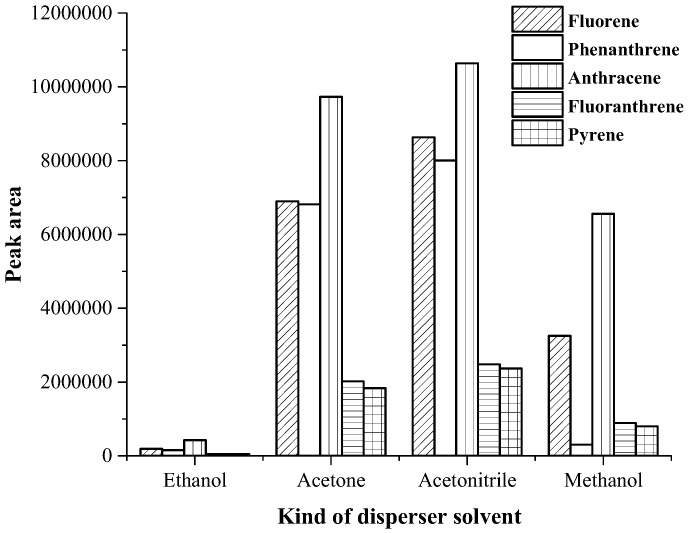
Effect of kind of disperser solvent on the extraction efficacy.

**Figure 4 toxics-07-00008-f004:**
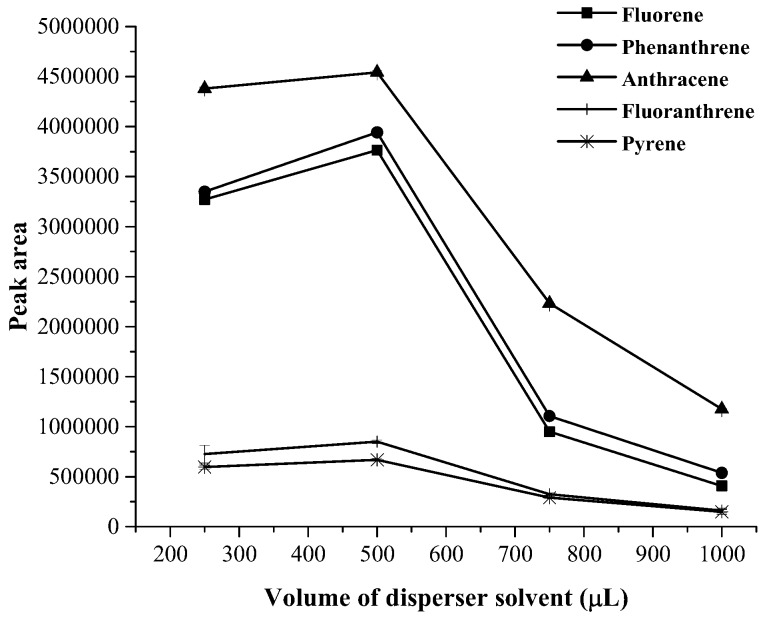
Effect of volume of disperser solvent on the extraction efficacy.

**Figure 5 toxics-07-00008-f005:**
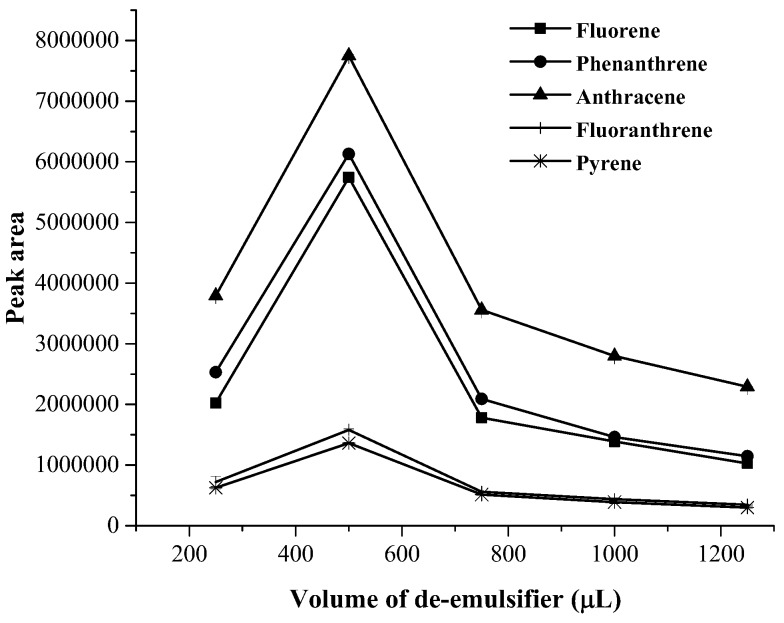
Effect of volume of de-emulsifier solvent on the extraction efficacy.

**Figure 6 toxics-07-00008-f006:**
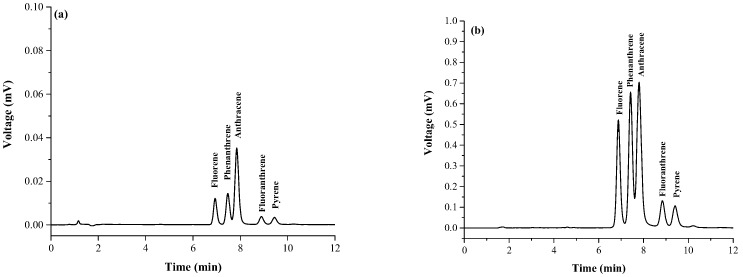
Chromatogram comparing analyses of (**a)** a mixture standard of polycyclic aromatic hydrocarbons with direct injection by HPLC and (**b**) a mixture of standard of polycyclic aromatic hydrocarbons after magnetic stirring assisted demulsified dispersive liquid–liquid microextraction technique: concentration of all standards was 100 ng·mL^−1^.

**Figure 7 toxics-07-00008-f007:**
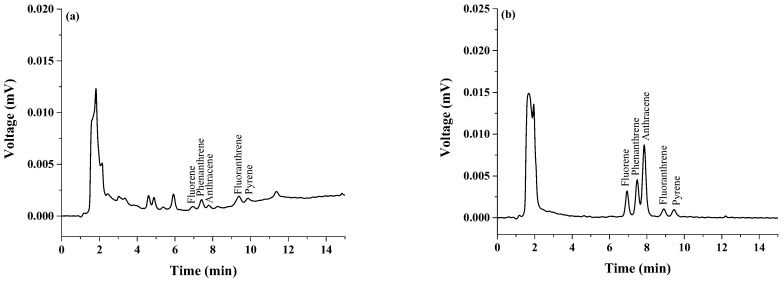
Chromatograms of (**a**) grilled pork sample, (**b**) grilled pork sample spiked with 0.01 mg·kg^−1^ of each polycyclic aromatic hydrocarbon, (**c**) grilled pork sample spiked with 0.05 mg·kg^−1^ of each polycyclic aromatic hydrocarbon, and (**d**) grilled pork sample spiked with 0.10 mg·kg^−1^ of each polycyclic aromatic hydrocarbon.

**Table 1 toxics-07-00008-t001:** Analytical performance of magnetic stirring assisted demulsified dispersive liquid–liquid microextraction method for determination of polycyclic aromatic hydrocarbons.

Analyte	Linear range (µg·mL^−1^)	Linear equation	*R* ^2^	EF	LOD (µg·mL^−1^)	LOQ (µg·mL^−1^)	Intra-day ^a^ (*n* = 5)		Inter-day (*n* = 3 × 5)	
							*t_R_*	Area	*t_R_*	Area
Fluorene	0.0005–1 (0.03–1) ^b^	*y* = (7 × 10^6^*x*) + 87,587	0.9992	65	0.0001	0.0003	0.15 (0.17)	2.18 (2.51)	0.39 (0.50)	6.72 (7.44)
Phenanthrene	0.0005–1 (0.03–1)	*y* = (7 × 10^6^*x*) + 87,587	0.9973	64	0.0001	0.0003	0.17 (0.19)	2.79 (3.08)	0.22 (0.29)	2.73 (3.19)
Anthracene	0.0005–1 (0.03–1)	*y* = 401,767*x* + 32,354	0.9993	60	0.0003	0.0005	0.24 (0.39)	2.58 (3.70)	0.37 (0.60)	5.68 (6.25)
Fluoranthrene	0.0005–1 (0.03–1)	*y*= (2 × 10^6^x) + 20,850	0.9989	66	0.0003	0.0005	0.20 (0.49)	3.55 (4.34)	0.33 (0.48)	5.51 (7.05)
Pyrene	0.0005–1 (0.03–1)	*y* = (4 × 10^6^*x*) + 30,813	0.9997	67	0.0001	0.0003	0.32 (1.06)	3.89 (5.40)	0.48 (1.48)	7.22 (8.12)

^a^ Precision was studied at standard concentration of 0.1 μg**·**mL^−1^; ^b^ the values in parentheses are gained from direct HPLC analysis.

**Table 2 toxics-07-00008-t002:** The determination of polycyclic aromatic hydrocarbons in the studied grilled pork samples (*n* = 3).

Samples	Amount found ± SD, mg·kg^−1^				
	Fluorene	Phenanthrene	Anthracene	Fluoranthrene	Pyrene
Grilled pork I (*n* = 3)	0.50 ± 0.10	1.00 ± 0.10	0.50 ± 0.10	0.80 ± 0.20	0.50 ± 0.01
Grilled pork II (*n* = 3)	-	0.70 ± 0.20	0.30 ± 0.20	0.50 ± 0.01	0.40 ± 0.01
Grilled pork III (*n* = 3)	0.30 ± 0.02	-	0.50 ± 0.20	0.70 ± 0.01	0.30 ± 0.20
Grilled pork IV (*n* = 3)	0.40 ± 0.10	0.40 ± 0.01	0.50 ± 0.10	0.60 ± 0.20	0.20 ± 0.10

-; not detected.

**Table 3 toxics-07-00008-t003:** Recovery obtained from the determination of polycyclic aromatic hydrocarbons in the studied grilled pork samples (*n* = 3).

Analytes	Spiked (mg·kg^−1^)	Grilled pork I		Grilled pork II		Grilled pork III		Grilled pork IV	
		RR (%)	RSD (%)	RR (%)	RSD (%)	RR (%)	RSD (%)	RR (%)	RSD (%)
Fluorene	0.01	83.45	1.52	87.64	4.73	95.73	6.63	91.57	8.76
	0.05	87.93	3.30	90.62	5.67	90.67	2.54	89.72	6.78
	0.10	84.63	6.35	96.87	3.79	85.33	2.63	98.74	6.87
Phenanthrene	0.01	82.93	5.36	92.39	4.33	87.73	3.87	89.30	7.45
	0.05	91.58	4.74	83.76	5.76	93.48	4.57	95.78	6.87
	0.10	98.74	6.79	90.78	7.63	90.63	7.86	90.87	7.86
Anthracene	0.01	82.97	4.53	87.33	3.63	92.78	6.75	90.63	3.67
	0.05	87.62	5.73	88.63	4.76	90.78	7.67	89.78	4.38
	0.10	90.73	8.63	93.74	6.78	91.87	1.56	89.90	3.78
Fluoranthrene	0.01	89.97	3.44	93.35	4.32	94.67	3.39	89.93	3.45
	0.05	91.56	4.67	89.90	6.32	91.75	4.13	83.63	6.78
	0.10	95.73	7.78	91.72	4.36	89.73	5.62	90.57	8.98
Pyrene	0.01	89.93	3.65	89.97	8.78	87.78	7.89	93.65	8.35
	0.05	90.67	4.78	91.63	5.67	88.98	6.78	91.32	5.56
	0.10	89.92	7.87	89.73	6.72	90.56	5.76	90.01	4.65

RR: Relative recovery; RSD: Relative standard deviation.

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
