# Peer review of "Magnetic Stirring Assisted Demulsification Dispersive Liquid–Liquid Microextraction for Preconcentration of Polycyclic Aromatic Hydrocarbons in Grilled Pork Samples"

_toxics, 2019, doi:10.3390/toxics7010008_

Reviewer 1 Report

Materials and methods

The names of countries and cities should be given for individual reagents and measuring equipment (e.g.Merck (Germany, city......));

In the same section authors should add section "method validation" and here, describe the validation parameters for the developed method and, above all, explain to the reader what is the Enrichment factors and how it is calculated.

Results and discussion

Section Analytical performance of the proposed method. Here the results of method validation should be described, and not all parameters of their calculation.
Validation parameters with the counting method, formulas should be found in the materials and methods section.

Figure 6. the names of the compounds on the chromatogram 2b should be poprwione because currently they are not readable

Figure 7. How authors explain chromatogram c and d.

Chromatogram c present grilled pork sample spiked with 0.05 mg/kg of each polycyclic aromatic hydrocarbons, chromatogram d sample spiked at the concnetration 0.1 mg/kg of PAHs but peaks are 5 and 10 times higher.

Author Response

Dear Reviewer I,

We thank you and the reviewers very much for the valuable comments.  The manuscript has been carefully revised accordingly to the comments as in the following: Manuscript entitled “Magnetic Stirring Assisted Demulsification Dispersive Liquid-Liquid Microextraction for Preconcentration of Polycyclic Aromatic Hydrocarbons in Grilled Pork Samples

All changes are highlighted in the revised manuscript using yellow colored.

Reviewer 2 Report

The paper is well structured, the abstract is concise and in the topic; the introduction is supported by well selected bibliographic data. The Experimental and Modeling Approach correctly. The manuscript is well written. Results and Discussions could be improved by studying other papers in the field:

Mohd Hassan FW et al., Dispersive liquid-liquid microextraction combined with dispersive solid-phase extraction for gas chromatography with mass spectrometry determination of polycyclic aromatic hydrocarbons in aqueous matrices. J Sep Sci. 2018 Oct;41(19):3751-3763. doi: 10.1002/jssc.201800326

Author Response

Dear Reviewer II,

We thank you and the reviewers very much for the valuable comments.  The manuscript has been carefully revised accordingly to the comments as in the following: Manuscript entitled “Magnetic Stirring Assisted Demulsification Dispersive Liquid-Liquid Microextraction for Preconcentration of Polycyclic Aromatic Hydrocarbons in Grilled Pork Samples

All changes are highlighted in the revised manuscript using yellow colored.
